# Strategies to improve retention in randomised trials: a Cochrane systematic review and meta-analysis

V C Brueton,[1] J F Tierney,[1] S Stenning,[1] S Meredith,[1] S Harding,[2] I Nazareth,[3] G Rait[3]

This article is based on a Cochrane Review published in the *Cochrane Database of Systematic Reviews* (CDSR) 2013, Issue 12, doi:10.1002/14651858.MR 000032.pub2 (see www.thecochranelibrary.com for information)

For numbered affiliations see end of article.

Correspondence to
Brueton VC;
v.brueton@ucl.ac.uk

## ABSTRACT

**Objective:** To quantify the effect of strategies to improve retention in randomised trials.

**Design:** Systematic review and meta-analysis.

**Data sources:** Sources searched: MEDLINE, EMBASE, PsycINFO, DARE, CENTRAL, CINAHL, C2-SPECTR, ERIC, PreMEDLINE, Cochrane Methodology Register, Current Controlled Trials metaRegister, WHO trials platform, Society for Clinical Trials (SCT) conference proceedings and a survey of all UK clinical trial research units.

**Review methods:** Included trials were randomised evaluations of strategies to improve retention embedded within host randomised trials. The primary outcome was retention of trial participants. Data from trials were pooled using the fixed-effect model. Subgroup analyses were used to explore the heterogeneity and to determine whether there were any differences in effect by the type of strategy.

**Results:** 38 retention trials were identified. Six broad types of strategies were evaluated. Strategies that increased postal questionnaire responses were: adding, that is, giving a monetary incentive (RR 1.18; 95% CI 1.09 to 1.28) and higher valued incentives (RR 1.12; 95% CI 1.04 to 1.22). Offering a monetary incentive, that is, an incentive given on receipt of a completed questionnaire, also increased electronic questionnaire response (RR 1.25; 95% CI 1.14 to 1.38). The evidence for shorter questionnaires (RR 1.04; 95% CI 1.00 to 1.08) and questionnaires relevant to the disease/condition (RR 1.07; 95% CI 1.01 to 1.14) is less clear. On the basis of the results of single trials, the following strategies appeared effective at increasing questionnaire response: recorded delivery of questionnaires (RR 2.08; 95% CI 1.11 to 3.87); a 'package' of postal communication strategies (RR 1.43; 95% CI 1.22 to 1.67) and an open trial design (RR 1.37; 95% CI 1.16 to 1.63). There is no good evidence that the following strategies impact on trial response/retention: adding a non-monetary incentive (RR=1.00; 95% CI 0.98 to 1.02); offering a non-monetary incentive (RR=0.99; 95% CI 0.95 to 1.03); 'enhanced' letters (RR=1.01; 95% CI 0.97 to 1.05); monetary incentives compared with offering prize draw entry (RR=1.04; 95% CI 0.91 to 1.19); priority postal delivery (RR=1.02; 95% CI 0.95 to 1.09); behavioural motivational strategies (RR=1.08; 95% CI 0.93 to 1.24); additional reminders to participants (RR=1.03;

## Strengths and limitations of this study

- This is the most comprehensive review of strategies specifically designed to improve retention in randomised trials, including many unpublished trials and data.
- Although our searches were extensive, some less well reported, ongoing, or unpublished trials, or trials conducted outside the UK might have been missed.
- Most of the evidence relates to increasing questionnaire response rather than ways to increase return of participants to sites.

95% CI 0.99 to 1.06) and questionnaire question order (RR=1.00, 0.97 to 1.02). Also based on single trials, these strategies do not appear effective: a telephone survey compared with a monetary incentive plus questionnaire (RR=1.08; 95% CI 0.94 to 1.24); offering a charity donation (RR=1.02, 95% CI 0.78 to 1.32); sending sites reminders (RR=0.96; 95% CI 0.83 to 1.11); sending questionnaires early (RR=1.10; 95% CI 0.96 to 1.26); longer and clearer questionnaires (RR=1.01, 0.95 to 1.07) and participant case management by trial assistants (RR=1.00; 95% CI 0.97 to 1.04).

**Conclusions:** Most of the trials evaluated questionnaire response rather than ways to improve participants return to site for follow-up. Monetary incentives and offers of monetary incentives increase postal and electronic questionnaire response. Some strategies need further evaluation. Application of these results would depend on trial context and follow-up procedures.

## INTRODUCTION

Loss of participants during trial follow-up can introduce bias and reduce power affecting the generalisability, validity and reliability of results.[1] [2] If losses are fewer than 5% they may lead to minimum bias, while 20% loss can threaten trial validity.[2] Missing data from losses to follow-up can be dealt with statistically; however, the risk of bias can remain.[3]

Trialists adopt various strategies to try to improve retention and generate maximum data return or compliance to follow-up procedures. These strategies are designed to motivate and keep participants or site clinicians engaged in a trial, but many are untested.[4] [5] A systematic review of strategies to retain participants in cohort studies suggests that providing incentives can improve retention.[6] Edwards' systematic review on methods to increase response rates to postal and electronic questionnaires across a range of study types found that including monetary incentives, keeping the questionnaire short and contacting people before questionnaires were sent were ways to increase response rates.[7] However, heterogeneity of effects was an issue and it is unclear which strategies are applicable to randomised trials. Moreover, reasons for loss to follow-up in cohort studies and surveys may differ from randomised trials. In trials, participants may be randomised to a study arm that is not their preferred choice and so strategies that improve retention in other study types cannot necessarily be extrapolated to randomised trials.

As loss to follow-up can compromise the validity of findings from randomised trials, delay results and potentially increase trial costs, we conducted a systematic review to assess the effect of strategies to improve retention in randomised trials.

## METHODS

The methods were prespecified in the Cochrane review protocol.[8]

### Trials included

We included randomised trials that compared strategies to increase participant retention embedded in 'host' randomised trials across disease areas and settings. These strategies should have been designed for use after participants were recruited and randomised. Retention trials embedded in cohort studies and surveys were excluded.

### Identification of retention trials

We searched MEDLINE (1950 to May 2012), EMBASE (1980 to May 2012), PsycINFO (1806 to May 2012), DARE (to May 2012), Cochrane CENTRAL and CINAHL (1981 to May 2012) using randomised controlled trial filters, where possible, and free text terms for retention (see online supplementary appendix 1 MEDLINE search). C2-SPECTR (to May 2009) and ERIC (1966 to May 2009) were only searched to May 2009 because of difficulties encountered with database and search platform changes. PreMEDLINE was searched to May 2009 but not subsequently because the free text records ultimately appear in MEDLINE. For search updates we also included the Cochrane Methodology Register, Current Controlled Trials metaRegister of Controlled Trials and WHO trials registry. Reference lists of relevant publications, reviews, included studies and abstracts of Society

for Clinical Trials meetings from 1980 to 2012 were also reviewed. No language restrictions were applied. All UK clinical trial units were surveyed to identify further eligible trials and the review was advertised at the Society for Clinical Trials Meeting in 2010.

### Trial selection

Two reviewers (VCB and GR) independently screened potentially eligible trials with disagreements resolved by a third author (SS). Information was sought from investigators to clarify eligibility where this was unclear.

### Data extraction

Data were extracted for each retention and host trial by one author (VCB) and checked by another (JFT). For retention trials, data were extracted on start time in relation to the host trial, aim, primary outcome, follow-up type, strategy to improve retention and comparator/s, including the frequency and time the strategy was administered, and numbers randomised, included and retained at the primary analysis. Data on sequence generation, allocation concealment, blinding and outcome reporting were extracted for each retention trial to assess risk of bias.[9] Data extracted for each host trial were: aim, comparators, primary outcome, disease area and setting. In addition, information on the sequence generation and allocation concealment was extracted to confirm that host trials were randomised. Missing or ambiguous data were queried or obtained through contact with trial authors.

### Statistical analysis

Retention was the primary outcome. Most of the retention strategies were applied during follow-up for the host trial. For three host trials, the retention strategy was applied in further follow-up of trial participants after completion. For four host trials, the strategy was applied during the pilot phase, and, for one other host trial, the retention strategy was applied before the host trial started. Where retention trials specified the primary outcome as the retention rate at a particular time point, this was used in the analysis. Where trials reported retention at multiple time points, without specifying which one was the primary outcome, we used the earliest time point in the analysis to see the initial impact on retention or response of introducing the strategy. Where trials reported time to retention, without specifying the primary time point, we used the final time point in the analysis, taking account of any censoring if data were available.

Retention trials with insufficient data could not be included in meta-analyses and were described qualitatively. Otherwise, risk ratios and their 95% CIs for retention were used to determine the effect of strategies on this outcome. The participant was the unit of analysis. Where clustering was ignored in the analysis of cluster randomised trials, we inflated the SEs using the intraclass correlation coefficients from appropriate external sources.[10–12]

For factorial trials[13] [14] that investigated different categories of strategies to improve retention, we included all trial comparisons in the relevant analyses and labelled these accordingly. For one factorial trial,[15] where the data were not available to perform this, only the broad trial comparisons (main effects) were included in the analyses. Where there were multiple comparisons in a single trial[16] within the same category of strategy, to avoid double counting, the intervention arms were combined and compared with the control arm. Similarly, for three-armed trials[17] [18] that compared two similar intervention arms with one control arm, the intervention arms were combined and compared with the control arm. For these trials, we also compared each intervention arm with the control arm, as separate trial comparisons, in exploratory analyses. Note that these approaches resulted in more trial comparisons than trials.

Heterogeneity was examined by the $\chi^2$ test, at 0.10 level of significance, and the $I^2$ statistic,[19] and explored through subgroup analyses. If there was no substantial heterogeneity, risk ratios were pooled using the fixed-effect model, but if heterogeneity was detected and was not explained by subgroup or sensitivity analyses, we did not pool the results. If heterogeneity could not be explained, we used the random effects model to assess the robustness of the results to the choice of model. To assess the robustness of the results, sensitivity analyses were conducted that excluded quasi-randomised trials.

The diversity of trials and interventions identified meant that not all of our prespecified subgroup analyses were appropriate or possible. Therefore, different types of strategies were analysed separately and new subgroups were defined within these strategies prior to analysis. These new analyses are listed in tables 1–4.

The absolute benefits of effective retention strategies were based on applying meta-analysis risk ratios to representative control arm retention rates.[20] All statistical analyses were conducted using RevMan5.

## RESULTS

We identified 38 eligible randomised retention trials from 24 304 records (see online supplementary figure S4). Retention trials were embedded in host trials, details of which are provided in the full Cochrane review.[12] Twenty-eight retention trials were published in full,[13–18 21–40] two in the grey literature[14 34] and eight are unpublished (*unpublished trials by Edwards, Svobodva, Letley, Maclennan, Land, Bailey 1, Bailey 2, Marson*). Unpublished trials were identified by word of mouth, reference lists of relevant literature and a survey of UK clinical trials units. Four retention trial publications contained two trials each.[18 32 33 35]

### Participants and settings

Eligible retention trials were from different geographical areas and clinical settings. Clinical areas ranged from exercise and alcohol dependency to treatment and screening for cancer (tables 1–4).[12]

Outcomes for strategies to improve retention were measured by: return of postal or electronic questionnaires[13–15 18 21 22 24 25 27 29–34 36–40] (*unpublished trials by Edwards, Svobodva, Letley, Maclennan, Land, Bailey 1, Bailey 2 Marson*) or biomedical data[17] (*Bailey 1, unpublished*) , a combination of postal, telephone and email follow-up[35] or face to face follow-up/retention.[16 28]

### Design of included retention trials

One retention trial was cluster randomised (*Land unpublished*), four were factorial trials[13–16] and there was one three-armed[17] and three four-armed trials.[18 32] Five trials were quasi-randomised,[16 28 29 33] allocating participants by either their identification numbers,[28 29] day of clinic visit[16] or by random selection of half the sample for the intervention and half for the control group.[33] All strategies targeted individual trial participants except one which targeted sites (*Land unpublished*).

Twenty-nine retention trials started during follow-up of the host trial[13 15 16 18 21 22 24–36 38 41] (*Edwards, Land, Maclennan, Bailey, Svoboda, unpublished*). One trial followed children of mothers who participated in the MRC ORACLE trial.[39] Two trials followed up participants in smoking cessation trials after the host trial finished.[17 40] Another retention trial randomised participants before the host trial started.[23] Four trials started during the pilot phase of the host trial[18 32 37] (*Letley unpublished*). For one trial, it is unclear when the retention trial started in relation to the host trial.[14]

### Incentive strategies

There were 14 retention trials of incentives and 19 trial comparisons. Thirteen trials investigating incentive strategies targeted questionnaire response, with only one targeting participant retention.[16] Incentive strategies aimed at improving questionnaire response were: vouchers,[18 29 39] cash,[25] a charity donation,[18] entry into a prize draw,[14 18 30] cheque[17] offers of study results[24 40] and a certificate of appreciation.[15 16] Incentive strategies aimed at participant retention were: lapel pins and a certificate of appreciation.[16] The UK incentives ranged in value from £5 to £20[18 29 39] (*Bailey 1, Baily 2, unpublished*) and from $2 to $10 for US-based trials, and were provided as either cash or voucher. Offers of entry into prize draws ranged from £25 to £250 for UK[18 30] and US $50 for US-based trials[14] (table 1); there was no information available on the chance of winning a prize. One trial evaluated giving a monetary incentive with a promise of a further incentive for return of trial data (*Bailey 2 unpublished*).

### Communication strategies

There were 14 retention trials of communication strategies and 20 trial comparisons. Most of the communication strategies targeted questionnaire response, with only one targeted at the return of biomedical test kits.[35]

**Table 1** Characteristics of included incentive trials

| Trial | Number randomised | Disease/ condition | Participant(s) | Setting | Intervention(s) | Control | Outcome retention trial | Time point used in analysis |
|---|---|---|---|---|---|---|---|---|
| *Addition of monetary incentive vs none* | | | | | | | | |
| Bauer 2004 (ab) | 300 | Treatment smoking dependence | Smokers | US community | (a) $10 cheque (b) $2 cheque Arms combined | No cheque | DNA specimen kit return plus postal questionnaire response | Overall number of kits returned |
| Gates 2009 | 2144 | Treatment neck injury | Patients with whiplash injury | UK hospital trusts | £5 voucher | No voucher | Postal questionnaire response at 2 weeks | 2 week response |
| Kenyon 2005 | 722 | Treatment preterm labour | Women 7 years post-participation in ORACLE trial | UK secondary care/ community | £5 voucher | No voucher | Postal questionnaire response | Overall response |
| *Addition of offer of monetary incentive/prize draw vs none* | | | | | | | | |
| Khadjesari 2011 (1ac) | 1022 | Treatment alcohol dependence | Adults scoring +5 on Audit C | UK Community: web based | (a) Offer £5 voucher, (c) Offer entry £250 prize draw Arms combined | No offer | Web-based questionnaire response | Response within 40 days of first reminder |
| Khadjesari 2011 (2) | 2591 | Treatment alcohol dependence | Adults scoring +5 on Audit C | Community: web based | Offer £10 Amazon voucher | No offer | Web-based questionnaire response | Response within 40 days of first reminder |
| *Addition of non-monetary incentive vs none* | | | | | | | | |
| Bowen 2000 (abc) | 4728 | Prevention lung cancer | Adults exposed to smoking and asbestos | US sites | (a) Certificate, (b) pin, (c) pin and certificate Arms combined | No certificate/ pin | Trial retention | Time from randomisation to first inactivation (stop taking vitamins or placebo) during PRIDE 2 year follow-up |
| Renfroe 2002 (a) | 664 | Treatment ventricular fibrillation ventricular tachycardia | Adults cardioverted from VT or resuscitated from VF | US hospital | Certificate of appreciation | No certificate | Postal questionnaire response | Overall response |
| Sharp 2006 (a) | 231 | Screening cervical cancer | Women with low-grade abnormal cervical smear | UK primary care | Pen | No pen | Postal questionnaire response | Overall response |
| Sharp 2006 (b) | 232 | Screening cervical cancer | Women with low-grade abnormal cervical smear | UK primary care | Pen | No pen | Postal questionnaire response | Overall response |

**Table 1** Continued

| Trial | Number randomised | Disease/ condition | Participant(s) | Setting | Intervention(s) | Control | Outcome retention trial | Time point used in analysis |
|---|---|---|---|---|---|---|---|---|
| Sharp 2006 (c) | 233 | Screening cervical cancer | Women with low-grade abnormal cervical smear | UK primary care | Pen | No pen | Postal questionnaire response | Overall response |
| Sharp 2006 (d) | 234 | Screening cervical cancer | Women with low-grade abnormal cervical smear | UK primary care | Pen | No pen | Postal questionnaire response | Overall response |
| *Addition of offer of non-monetary incentive vs no offer* | | | | | | | | |
| Cockayne 2005 | 1038 | Prevention fracture | Women with hip fracture risk factors | UK primary care | Offer of study results | No offer | Postal questionnaire response | Overall response |
| Hughes 1989 | 100 | Treatment smoking dependence | Adult smokers | US community | Offer results reprint | No offer | Postal questionnaire response | Overall response |
| *Addition of offer of monetary donation to charity vs no offer* | | | | | | | | |
| Khadjesari 2011 (1b) | 815 | Treatment alcohol dependence | Adults scoring +5 on Audit C | Community: on line | Offer £5 charity donation | No offer | Web-based questionnaire response | Response within 40 days of first reminder |
| *Addition of £10 plus offer of £10 vs addition of £5 plus offer of £5* | | | | | | | | |
| Bailey (2) (unpublished) | 417 | Promotion sexual health | Young people | Community UK on line | Offer of £20 shopping voucher | Offer of £10 shopping voucher | Postal questionnaire response | Response at 3-month follow-up |
| *Addition of £20 voucher offer vs addition of £10 voucher offer* | | | | | | | | |
| Bailey (1) (unpublished) | 485 | Promotion of sexual health | Young people | Community UK on line | £10 shopping voucher + offer of £10 shopping voucher | £5 shopping voucher + offer of £5 shopping voucher | Postal questionnaire response and chlamydia kit return | Response at 3-month follow-up |
| *Addition of monetary incentive vs offer of entry into prize draw* | | | | | | | | |
| Kenton 2007 (a) | 147 | Prevention of postnatal depression | Women postpartum at high risk of postnatal depression | Canada community | $2 coin | Draw for $50 gift voucher | Postal questionnaire response | Overall response |
| Kenton 2007 (b) | 150 | Prevention of postnatal depression | Women postpartum at high risk of postnatal depression | Canada community | $2 coin | Draw for $50 gift voucher | Postal questionnaire response | Overall response |
| *Offer of prize draw entry vs no offer* | | | | | | | | |
| Leigh Brown 1997 | 1307 | Clinical management of orthopaedic conditions | Adults non-surgical musculoskeletal conditions | UK hospital out patients department | Aware offer of monthly prize draw of £25 gift voucher | No offer | Postal questionnaire response after 1st and 2nd reminder | No data available |

**Table 2** Characteristics of included communication trials

| Trial | Number randomised | Disease/condition | Participants | Setting | Intervention(s) | Control | Outcome retention trial | Time point used in analysis |
|---|---|---|---|---|---|---|---|---|
| *Enhanced letter vs standard letter* | | | | | | | | |
| Renfroe 2002 (c) | 664 | Treatment of ventricular tachycardia (VT) ventricular fibrillation (VF) | Adults cardioverted from VT or resuscitated from VF | US hospital | Cover letter signed by physician | Cover letter signed by coordinator | Postal questionnaire response | Overall response |
| Marson 2007 | 1815 | Treatment of epilepsy | Adults with epilepsy | UK hospital outpatient departments | Letter explaining the approximate time needed to complete the questionnaire | Standard letter | Postal questionnaire response | Overall response |
| *Total design postal method for postal questionnaires vs customary method* | | | | | | | | |
| Sutherland 1996 | 226 | Prevention of breast cancer | Women with 50% breast volume dysplasia | Canada hospital clinic | Total design method for postal follow-up | Customary method for postal follow-up | Postal questionnaire response | Response at day 70 |
| *Priority vs regular post* | | | | | | | | |
| Renfroe 2002 (b) | 664 | Treatment of ventricular tachycardia (VT) ventricular fibrillation (VF) | Adults cardioverted from VT or resuscitated from VF | US hospital | Overnight questionnaire delivery | Standard questionnaire delivery | Postal questionnaire response | Overall response number of questionnaires returned |
| Sharp 2006 (e) | 233 | Screening of cervical cancer | Women with low-grade abnormal cervical smear | UK primary care | First class outward post | Second class outward post | Postal questionnaire response | Overall response |
| Sharp 2006 (f) | 231 | Screening of cervical cancer | Women with low-grade abnormal cervical smear | UK primary care | First class outward post | Second class outward post | Postal questionnaire response | Overall response |
| Sharp 2006 (g) | 240 | Screening of cervical cancer | Women with low-grade abnormal cervical smear | UK primary care | Stamped reply envelope | Business reply envelope | Postal questionnaire response | Overall response |
| Sharp 2006 (h) | 223 | Screening of cervical cancer | Women with low-grade abnormal cervical smear | UK primary care | Stamped reply envelope | Business reply envelope | Postal questionnaire response | Overall response |

Continued

**Table 2**  Continued

| Trial | Number randomised | Disease/ condition | Participants | Setting | Intervention(s) | Control | Outcome retention trial | Time point used in analysis |
|---|---|---|---|---|---|---|---|---|
| Kenton 2007 (c) | 149 | Screening of postnatal depression | Women postpartum at high risk of postnatal depression | Canada community | Priority outward mail | Regular outward mail | Postal questionnaire response | Overall response |
| Kenton 2007 (d) | 148 | Screening of postnatal depression | Women postpartum at high risk of postnatal depression | Canada community | Priority outward mail | Regular outward mail | Postal questionnaire response | Overall response |
| *Additional reminder vs usual follow-up procedures* | | | | | | | | |
| Ashby 2011 | 148 | Prevention of migraine | Adults with a history of two migraine attacks | UK community | Electronic reminder (email and/or SMS text) | No electronic reminder | Postal questionnaire response | Response at 40 days |
| Maclennan unpublished | 753 | Prevention of fracture | Adults with a history of osteoporotic fracture | UK hospital | Telephone reminder (before receiving first reminder) | No telephone reminder | Postal questionnaire response | Overall response response rate |
| Nakash unpublished | 298 | Treatment of ankle injury | Adults with acute severe ankle sprain | UK accident and emergency departments | Trial calendar with questionnaire due dates | No calendar | Postal questionnaire response at 4, 12 weeks, and 9 months | Response at 4 weeks |
| Severi 2011 (1) | 1950 | Treatment of smoking dependence | Adult smokers willing to quit in Txt2stop | UK community | Text message and fridge magnet emphasising social benefits of study participation | Text message 3 days after questionnaire sent reminding questionnaire is due | Postal questionnaire response | Response at 30 weeks from randomisation |
| Severi 2011 (2) | 127 | Treatment of smoking dependence | Adult smokers willing to quit in Txt2stop | UK community | Telephone reminder from principal investigator that participant is 6 weeks overdue returning their specimen | Standard text and no phone call from principle investigator | Return of cotinine samples | Completed cotinine sample follow-up for Txt2stop at end of May 2009 |

Continued

**Table 2** Continued

| Trial | Number randomised | Disease/ condition | Participants | Setting | Intervention(s) | Control | Outcome retention trial | Time point used in analysis |
|---|---|---|---|---|---|---|---|---|
| Man 2011 | 125 | Treatment of back pain | Adults with back pain | UK primary care | SMS text reminder message as follow-up questionnaire sent out | No SMS text message | Postal questionnaire response | Overall response rate |
| *Monthly reminder of upcoming assessment to site vs usual reminders* | | | | | | | | |
| Land 2007 | 429 | Treatment of breast cancer | Women with ductal carcinoma in situ | Hospital sites USA, Canada, Puerto Rico | Prospective monthly reminder of upcoming assessments to sites | No extra reminders to sites | Postal questionnaire response | Overall response rate |
| *Early vs late administartion of questionnaire* | | | | | | | | |
| Renfroe 2002 (d) | 664 | Treatment of ventricular tachycardia (VT) ventricular fibrillation (VF) | Adults cardioverted from VT or resuscitated from VF | US hospital | Questionnaire sent 2–3 weeks after last AVID follow-up visit | Questionnaire sent 1-4 months after last AVID follow-up visit | Postal questionnaire response | Overall response number of questionnaires returned |
| *Recorded delivery vs telephone reminder* | | | | | | | | |
| Tai 1997 | 192 | Clinical management of asthma and diabetes | Adults with asthma or diabetes | UK primary care | Recorded delivery reminder | Telephone reminder | Postal questionnaire response | Overall response number of questionnaires returned |
| *Telephone interview vs questionnaire and monetary incentive* | | | | | | | | |
| Couper 2007 | 700 | Weight management | Adults with BMI >25 | US community web based | Telephone interview by trained interviewer | Postal questionnaires with $5 bill | Post and telephone questionnaire response | Response at 6 months |

**Table 3** Characteristics of included trials evaluating new questionnaire strategies

| Trial | Number randomised | Disease/ condition | Participants | Setting | Intervention(s) | Control | Outcome retention trial | Time point used in analysis |
|---|---|---|---|---|---|---|---|---|
| *Short vs long questionnaire* | | | | | | | | |
| Dorman 1997 | 2253 | Treatment of stroke | Patients with stroke | UK hospital | Short EUROQOL questionnaire | Long SF 36 questionnaire | Postal questionnaire response after first mail out and reminder | Response at first time point |
| Edwards 2001 unpublished | 99 | Treatment of head injury | Head injury patients | UK hospital intensive care units | 1-page, 7 question functional dependence questionnaire | 3-page, 16 question functional dependence questionnaire. | Postal questionnaire response | Response at 3 months |
| Svoboda 2001 unpublished | 91 | Treatment of head injury | Head injury patients | Czech republic hospital intensive care units | 1-page, 7 question functional dependence questionnaire | 3-page, 16 question functional dependence questionnaire. | Postal questionnaire response | Response at 3 months |
| McCambridge 2011 1b | 2835 | Treatment of alcohol dependence | Adults scoring +5 on Audit C | Community web based | Audit Short (alcohol use disorders questionnaire) + LDQ (Leeds dependancy questionnaire) | APQ (alcohol problems questionnaire) | Web-based questionnaire response at 1 month and 3 months | Response at 1 month |
| McCambridge 2011 2b | 1999 | Treatment of alcohol dependence | Adults scoring +5 on Audit C | Community web based | Audit Short (alcohol use disorders questionnaire) + LDQ (Leeds dependancy questionnaire) | APQ (alcohol problems questionnaire) | Web-based questionnaire response at 3 month and 12 months | Response at 3 months |
| *Long and clear vs short and condensed questionnaires* | | | | | | | | |
| Subar 2001 | 900 | Screening prostate, lung, ovarian, colorectal cancer | Adults in PLCO trial | US sites | DHQ (36-page food frequency questionnaire) | PLCO (16-page food frequency questionnaire) | Postal questionnaire/ response on site completion | Overall response |
| *Question order: condition first vs generic first questions* | | | | | | | | |
| McColl 2003 (1) | 4751 | Clinical management of asthma | Adults with asthma | UK primary care | Condition-specific questions first followed by generic | Generic questions followed by condition specific | Postal questionnaire response | Overall response |
| McColl 2003 (2) | 4684 | Clinical management of angina | Adults with angina | UK primary care | Condition-specific questions followed by generic | Generic questions followed by condition specific | Postal questionnaire response | Overall response |

Continued

**Table 3**  Continued

| Trial | Number randomised | Disease/condition | Participants | Setting | Intervention(s) | Control | Outcome retention trial | Time point used in analysis |
|---|---|---|---|---|---|---|---|---|
| Letley unpublished. No data available | Data not available | Treatment of back pain | Adults with low back pain | UK primary care | 23-page self-completion Roland disability questionnaire at front and SF 36 at back | vice versa | Questionnaire response | No data |
| *Questionnaire: relevant vs less relevant to condition* | | | | | | | | |
| McCambridge 2011 1a | 1892 | Treatment of alcohol dependence | Adults scoring +5 on Audit C | Community web based | Alcohol problem questionnaire (APQ)23 items | Core OM Mental health assessment 23/34 items | Web-based questionnaire response at 1 and 3 months | Response at 1 month |
| McCambridge 2011 2a | 2001 | Treatment of alcohol dependence | Adults scoring +5 on Audit C | Community web based | Audit Short (alcohol use disorders questionnaire) + LDQ (Leeds dependancy questionnaire) | Core OM Mental health assessment 10 items | Web-based questionnaire response at 3 and 12 months | Response at 3 months |

**Table 4**  Characteristics of other trials

| Trial | Number randomised | Disease/condition | Participants | Setting | Intervention(s) | Control | Outcome retention trial | Time point used in analysis |
|---|---|---|---|---|---|---|---|---|
| *Motivation vs information* | | | | | | | | |
| Cox 2008 | 120 | Exercise improvement | Sedentary Women | Australia Community | Motivational workshops and newsletters | Information sheets and newsletters | Program and trial retention at 6 and 12 months | 6-month and 12-month data. Data for 6 months used |
| Chaffin 2009 | 153 | Parenting improvement | Adults referred for parenting improvement | US community | Self-motivation information | Standard information | Program attendance/ trial retention | Retention at 12 weeks |
| *Case management vs usual follow-up* | | | | | | | | |
| Ford 2006 | 703 | Screening prostate, lung, ovarian, colorectal cancer | Adults in the PLCO screening trial | US sites | In-depth case management | Regular trial procedures | Attendance at face to face cancer screening | Retention at 3 years |
| *Open vs blind trial design* | | | | | | | | |
| Avenell 2004 | 538 | Prevention of fracture | Adults with a history of osteoporotic fracture | UK hospital | Open trial design | Blind trial design | Postal questionnaire response at 4, 8, 12 months | Response at 12 months |

Strategies evaluated were: enhanced letters, that is, those with additional information about trial processes or with an extra feature, for example, signed by a principal investigator[15] (*Marson unpublished*), use of additional telephone reminders[35] (*Maclennan unpublished*), a calendar including reminders of when to return a questionnaire[34], text and/or email reminders[21] [31] [35] and reminders to sites of upcoming assessments versus no additional reminder (*Land unpublished*). One trial used a package of postal communication strategies called the Total Design Method (TDM)[37] and another used recorded delivery of questionnaires[38] (table 2).

Five trials evaluated communication and incentive strategies[13–15] [25] [35] (tables 1 and 2). The incentives were: certificates of appreciation for study involvement,[15] study-branded pens,[13] a US$2 coin[14] and a US$5 bill[25] or fridge magnets.[35] The communication strategies were: first or second class outward post,[13–15] stamped and business reply envelopes,[13] letters signed by different study personnel,[15] letters posted at different times,[15] telephone survey[25] and text messages.[35]

### New questionnaire formats

The effect of a change in questionnaire format on response to questionnaires was evaluated in eight trials. The 10 comparison formats evaluated were (table 3): questionnaire length[27] [32] [36] (*Edwards unpublished Svoboda unpublished*), order of questions (*Letley unpublished*[33]) and relevance of questionnaires in the context of research in alcohol dependence[32].

### Behavioural strategies

There were two retention trials of motivational behavioural strategies, one in an exercise trial[26] and another in a parenting trial[23] (table 4). A behavioural strategy was defined as giving participants information about goal setting and time management to facilitate successful trial completion. One retention trial was run prior to the host trial,[23] where only participants who completed the orientation/retention trial were included in the subsequent parenting trial.

### Case management

Case management defined as outreach, service planning linkage, monitoring and advocacy was compared with usual follow-up in a cancer screening trial[28] (table 4). This strategy involved trial assistants managing participant follow-up by arranging services to enable participants to keep trial follow-up appointments.

### Methodology strategies

One trial included an open trial versus blind trial design to evaluate the impact on questionnaire response[22] (table 4).

### Trials not included in the meta-analyses

Two included trials could not be included in the meta-analysis[30] (*Letley unpublished*). For one trial, the

host trial participants included randomised and non-randomised participants[30] and the author confirmed that the participants in the retention trial were from both cohorts and these data could not be separated. For the other trial, retention trial outcome data were not available (*Letley unpublished*).

### Risk of bias in included trials

Twenty-four trials describe adequate sequence generation[15] [16] [18] [22–24] [26] [30–32] [34] [35] [37] [39] [40] (*unpublished trials Bailey 1, Bailey 2, Letley, Land, Maclennan, Marson*). There was insufficient information about the sequence generation for 10 trials, but they were all described as randomised[13] [14] [17] [21] [25] [27] [36] [38] (*Edwards, Svoboda unpublished*). Five trials used quasi-randomisation.[16] [28] [29] [33] Fifteen trials reported adequate sequence generation and allocation concealment[18] [22] [24] [26] [31] [32] [34] [39] [40] (*Letley, Maclennan, Bailey 1, Bailey 2, unpublished*).

Blinding of participants to the intervention was not possible for incentive strategies, offers of incentives, behavioural or case management strategies and different types of communication and questionnaire format strategies. For one trial that evaluated the effect of a blind versus open design on retention this was not applicable.[22] For some trials, participants were aware of the intervention but unaware of the evaluation[14] [16] [23] [30] [33] [39] (*Maclennan, Marson unpublished*). For another trial,[26] exercise sessions were not separated according to the behavioural intervention, that is, walking and swimming, and potential contamination between groups could have led to bias. For other trials, blinding of participants or trial personnel to the outcome or intervention was not reported. The primary outcome measure for this review was retention, and this was well reported. Authors were contacted for clarification of any exclusions after randomisation if this was unclear from retention trial reports. Although retention trial protocols were not available for included trials, the published and unpublished reports included reported all expected outcomes for retention.

### The effects of strategies

#### Incentive strategies

There were 14 retention trials of incentives, 19 trial comparisons with 16 253 comparisons. Across incentive subgroups, there was considerable heterogeneity (p<0.00001; figure 1A). So we did not pool the results for incentives. Unless otherwise stated, results from the random effects model were similar. Three trials (3166 participants) that evaluated the effect of giving monetary incentives to participants showed that the addition of monetary incentives is more effective than no incentive at increasing response to postal questionnaires (RR=1.18; 95% CI 1.09 to 1.28; p<0.0001, heterogeneity p=0.21; figure 1A). A sensitivity analysis excluding the quasi-randomised trial by Gates *et al*[29] shows a similar effect (RR=1.31; 95% CI 1.11 to 1.55; p=0.002). Also, based on two web-based trials (3613 participants, figure 1A), an offer of a monetary incentive

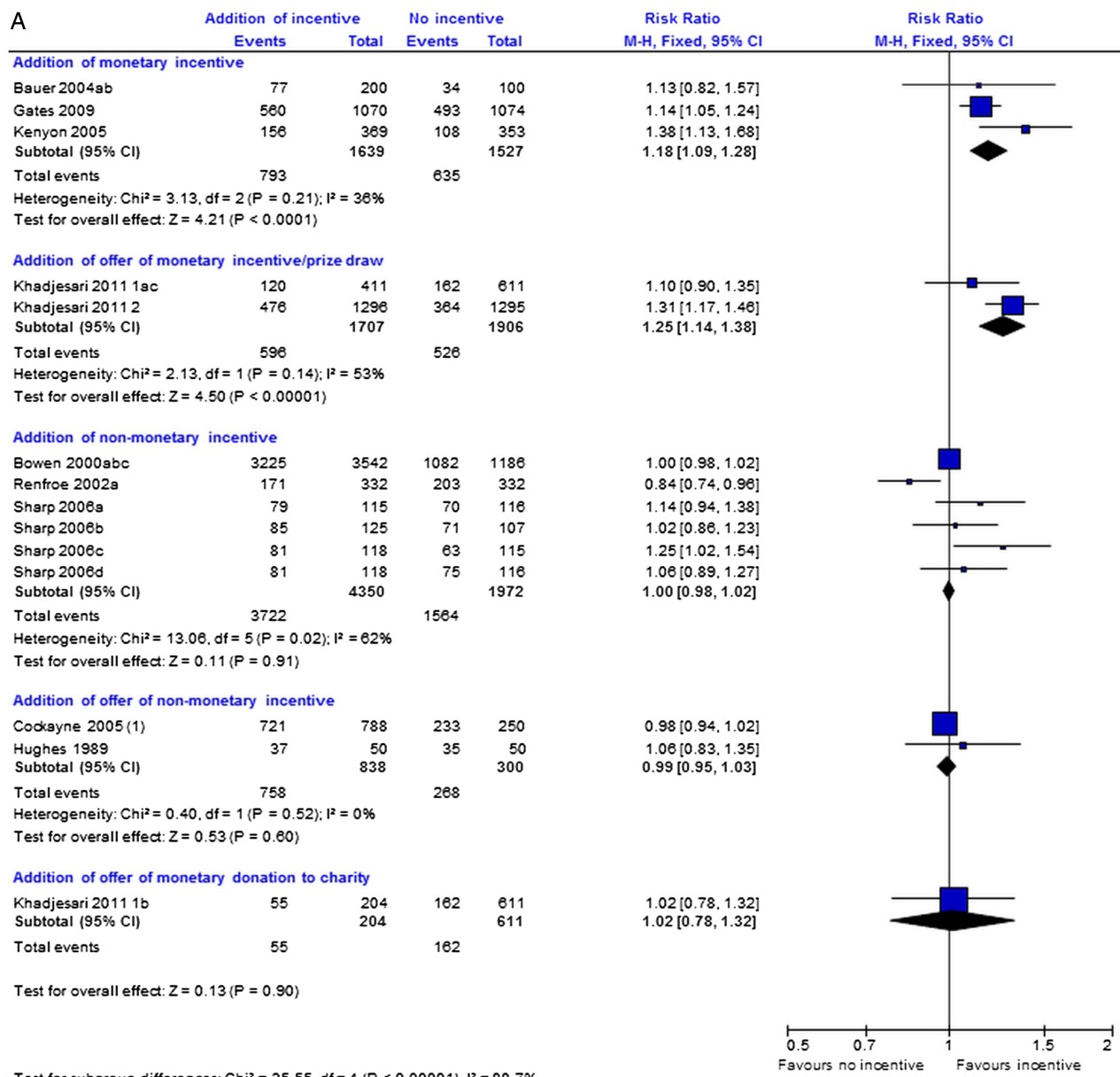

**Figure 1**  (A) Incentive strategies: main analysis addition of incentive versus no incentive, (B) incentives: addition of £20 vs £10 incentive, (C) incentives addition of: monetary incentive versus offer of entry into prize draw.

promotes greater return of electronic questionnaires than no offer (RR=1.25; 95% CI 1.14 to 1.38, p<0.00001, heterogeneity p=0.14). However, a single trial comparison suggests that an offer of a monetary donation to charity does not increase response to electronic questionnaires (RR=1.02, 95% CI 0.78 to 1.32; p=0.90; figure 1A).

On the basis of three trials (6322 participants), there is no clear evidence that the addition of non-monetary incentives improved questionnaire response (RR=1.00; 95% CI 0.98 to 1.02; p=0.91) but there is some heterogeneity (p=0.02; figure 1A). A sensitivity analysis excluding the quasi-randomised trial by Bowen et al[16] showed a similar effect (RR=1.00; 95% CI 0.93 to 1.08; p=0.99, heterogeneity p=0.01). Two trials (1138 participants) evaluating offers of non-monetary incentives suggest that an

offer of a non-monetary incentive is neither more nor less effective than no offer (RR=0.99; 95% CI 0.95 to 1.03; p=0.60; heterogeneity p=0.52) at improving questionnaire response (figure 1A).

In exploratory analyses, the different incentive arms that were combined for the main analysis do not appear to show differential effects (see online supplementary figure S5).

Two trials (902 participants) show that higher value incentives are better at increasing response to postal questionnaires than lower value incentives (RR 1.12; 95% CI 1.04 to 1.22; p=0.005; heterogeneity p=0.39) irrespective of how they are given (figure 1B).

Two trial comparisons (297 participants) provide no clear evidence that giving a monetary incentive is better than an offer of entry into a prize draw for improving

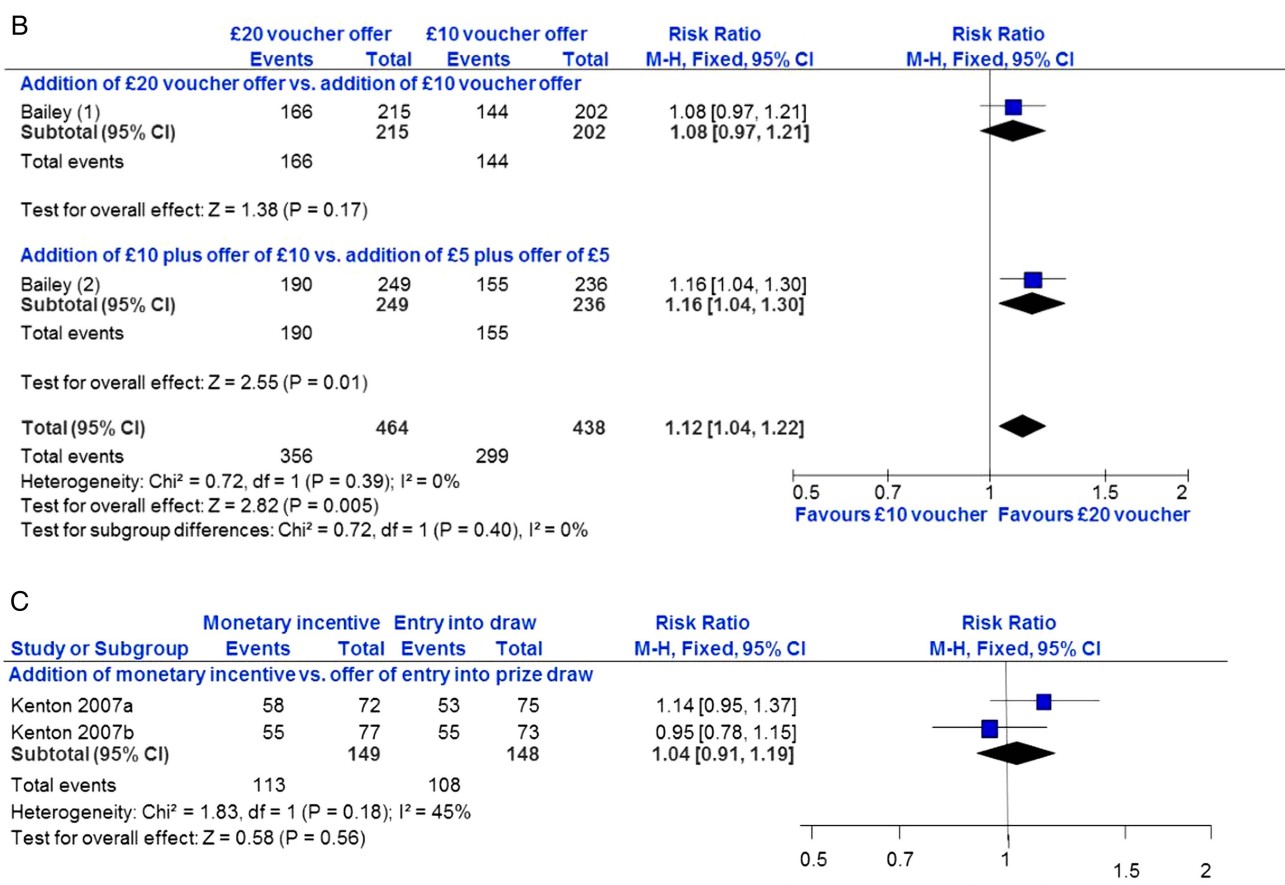

**Figure 1** Continued.

response to postal questionnaires (RR=1.04; 95% CI 0.91 to 1.19; p=0.56, heterogeneity p=0.18; figure 1C).

One trial could not be included in the analysis,[30] but showed a higher response in the group offered entry into a prize draw (70.5%) compared with the group not offered entry into the draw (65.8%).

### Communication strategies

There were 14 trials of communication strategies and 20 comparisons with 9822 participants. The communication strategies were so diverse that these were analysed separately.

Results from two trials (2479 participants) show that an enhanced letter is neither more nor less effective than a standard letter for increasing response to postal questionnaires (RR=1.01; 95% CI 0.97 to 1.05; p=0.70; heterogeneity p=0.80; figure 2A). Although based on a single trial (226 participants), the TDM package seems much more effective than a customary postal communication method at increasing questionnaire return (RR=1.43, 95% CI 1.22 to 1.67; p<0.0001; figure 2B). Based on the relevant arms of three trials (1888 participants), there is no clear evidence that priority post is either more or less effective than regular post at increasing trial questionnaire return (RR=1.02; 95% CI 0.95 to 1.09; p=0.55; heterogeneity p=0.53; figure 2C).

Six trials (3401 participants) evaluated the effect of different types of reminders to participants on questionnaire response. There is no clear evidence that a reminder is either more or less effective than no reminder (RR=1.03; 95% CI 0.99 to 1.06; p=0.13; heterogeneity p=0.73) at improving trial questionnaire response (figure 2D). One trial (700 participants) showed no clear evidence that a telephone survey is either more or less effective than a monetary incentive and a questionnaire for improving questionnaire response (RR=1.08; 95% CI 0.94 to 1.24; p=0.27; figure 2E). Based on one cluster randomised trial (272 participants), a monthly reminder to sites of upcoming assessment was neither more nor less effective than the usual follow-up (RR=0.96; 95% CI 0.83 to 1.11; p=0.57). However, one small trial (192 participants) suggested that recorded delivery is more effective than a telephone reminder (RR=2.08; 95% CI 1.11 to 3.87; p=0.02). Based on one other trial (664 participants), there is no clear evidence that sending questionnaires early increased or decreased response (RR=1.10; 95% CI 0.96 to 1.26; p=0.19).

### New questionnaire strategies

Eight trials with 10 comparisons (21 505 participants) evaluated the effect of a new questionnaire format on questionnaire response. Although there is only some

**Figure 2** Communication
strategies: (A) enhanced versus
standard letter, (B) total design
versus customary post, (C)
priority versus regular post, (D)
additional reminders to
participants versus usual
follow-up, (E) telephone survey
versus monetary incentive and
questionnaire.

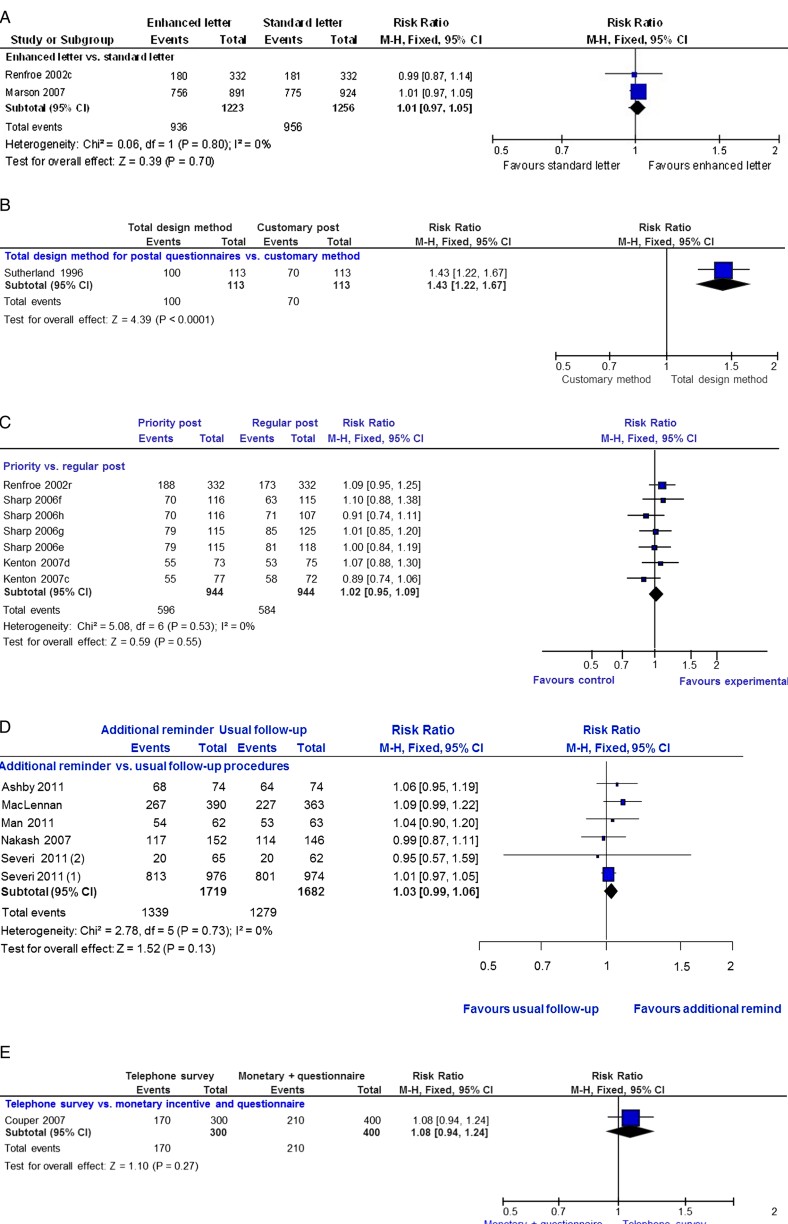

heterogeneity between the questionnaire subgroups (p=0.11; figure 3), it did not seem reasonable to pool the results based on such different interventions.

Five trials (7277 participants) compared the effect of short versus long questionnaires on postal questionnaire response. There is only a suggestion that short questionnaires may be better (RR=1.04; 95% CI 1.00 to 1.08; p=0.07, heterogeneity p=0.14; figure 3). Based on one trial (900 participants), there is no clear evidence that long and clear questionnaires are more or less effective than shorter condensed questionnaires for increasing questionnaire response (RR=1.01, 0.95–1.07; p=0.86; figure 3). Two quasi-randomised trials (9435 participants) also show no good evidence that placing disease/condition questions before generic questions is either more or less effective than vice versa at increasing questionnaire response (RR=1.00, 0.97–1.02; p=0.75, heterogeneity

p=0.44; figure 3). One trial by Letley (*unpublished*), not included in this analysis, provided no estimate of effect.

In the context of research on reducing alcohol consumption, there is also evidence that more relevant questionnaires, that is, those relating to alcohol use, increase response rates (RR 1.07; 95% CI 1.01 to 1.14; p=0.03, figure 3).

### Behavioural/motivational strategies

Two community-based trials (273 participants) show no clear evidence that the behavioural/motivational strategies used are either more or less effective than standard information for retaining participants (RR=1.08; 95% CI 0.93 to 1.24; p=0.31, heterogeneity p=0.93).

### Case management strategies

One trial (703 participants) evaluated the effect of intensive case management procedures on retention.

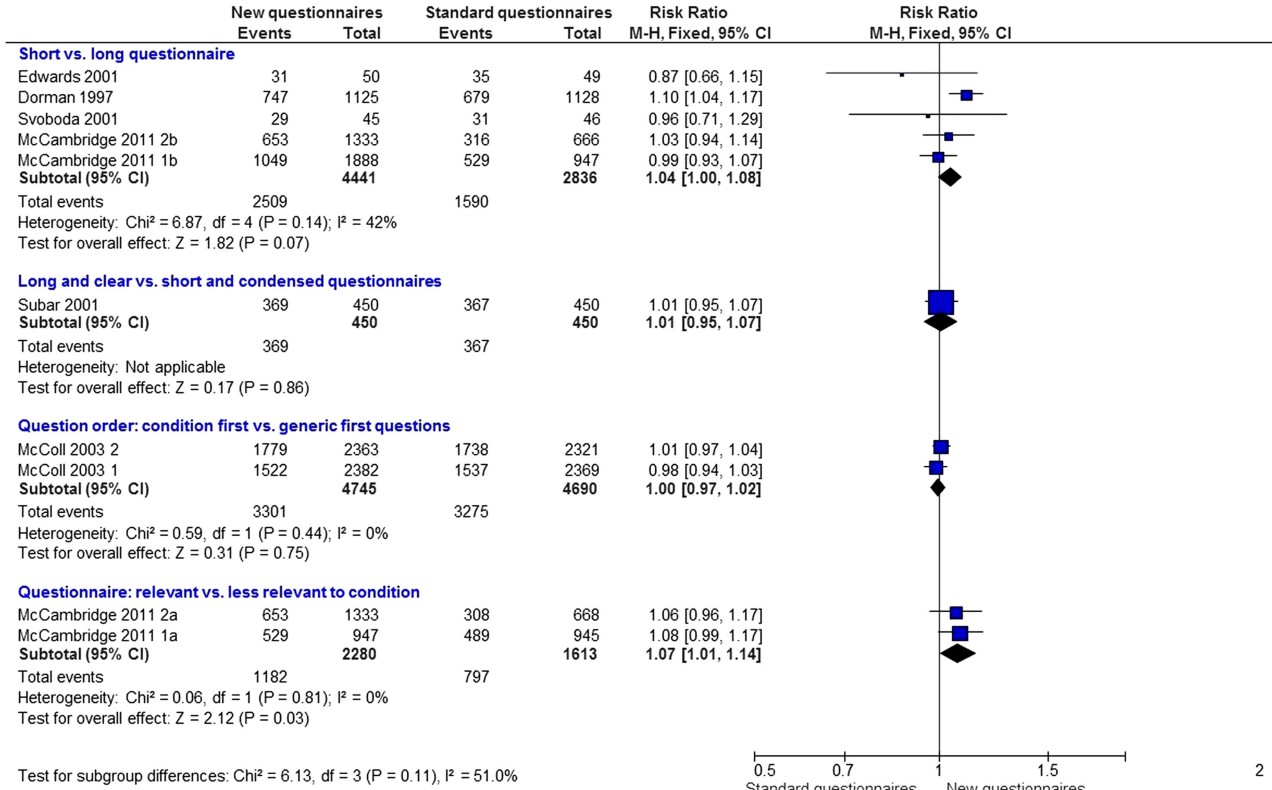

**Figure 3** Questionnaires: new format versus standard format.

There is no evidence that intensive case management is either more or less effective than usual follow-up in the population examined (RR=1.00; 95% CI 0.97 to 1.04; p=0.99).

## Methodology strategies

One fracture prevention trial (538 participants) evaluated the effect of participants knowing their treatment allocation (open trial) compared with participants blind/unaware of their allocation on questionnaire response. The open design led to higher response rates (RR=1.37; 95% CI 1.16 to 1.63; p=0.0003).

### Absolute benefits of strategies to improve retention

The absolute benefits of effective strategies on typical questionnaire response are illustrated in table 5. Based on a 40% baseline response rate for postal questionnaires, the addition of a monetary incentive is estimated

to increase response by 92 questionnaires/1000 sent (95% CI 50 to 131). With a baseline response rate of 30%, as seen in the included online trial, the addition of an offer of a monetary incentive is estimated to increase response by 140 questionnaires/1000 sent (95% CI 86 to 193).

## DISCUSSION

Thirty-eight randomised retention trials were included in this review, evaluating six broad types of strategies to increase questionnaire response and retention in randomised trials. Trials were conducted across a spectrum of disease areas, countries, healthcare and community settings (tables 1–4). Strategies with the clearest impact on questionnaire response were: addition of monetary incentives compared with no incentive for return of postal questionnaires, addition of an offer of a monetary incentive when compared with none for return of

| Table 5 Absolute benefit of effective strategies to improve retention | | | | | | | | | |
|---|---|---|---|---|---|---|---|---|---|
| **Example of proportion of questionnaires returned in control arm** | | | 30% | 40% | 50% | 60% | 70% | 80% | 90% |
| **Strategy to improve retention** | **RR** | **1/RR** | | | | | | | |
| Addition of monetary incentive vs no incentive | 1.18 | 0.847 | 107 | 92 | 76 | 61 | 5 | 3 | 2 |
| Addition of offer of monetary incentive/prize draw vs no offer | 1.25 | 0.800 | 140 | 120 | 100 | 80 | 60 | 40 | 20 |
| Addition of higher value monetary incentive vs addition of lower amount | 1.12 | 0.890 | 77 | 66 | 55 | 44 | 33 | 22 | 11 |

electronic questionnaires and an offer of £20 vouchers when compared with £10 for return of postal questionnaires and biomedical test kits. The evidence was less clear about the effect of shorter questionnaires rather than longer questionnaires and for questionnaires of greater relevance to the questions being studied. Recorded delivery of questionnaires, the TDM, a 'package' of postal communication strategies with reminder letters and an open trial design appear more effective than standard procedures. These strategies were tested in single trials and may need further evaluation. The addition of a non-monetary incentive or an offer of a non-monetary incentive compared with no incentive did not increase or decrease trial questionnaire response. 'Enhanced' letters, letters delivered by priority post or additional reminders were also no more effective than standard communication. Altering questionnaire structure does not seem to increase response. No strategy had a clear impact on increasing the number of participants returning to sites for follow-up.

## Strengths and weaknesses

This is the most comprehensive review of strategies specifically designed to improve retention in randomised trials, including many unpublished trials and data. Although our searches were extensive, some less well-reported, ongoing, or unpublished trials, or trials conducted outside the UK might have been missed.

Most of the trials used appropriate methods for randomisation or at least stated that they were randomised. For trials that did not describe their methods well or provide further information, there remains a potential risk of selection bias. Sensitivity analyses excluding quasi-randomised trials did not affect the results. In this context, where motivating participants to provide data or attend clinics is often the target of the interventions and so appropriately influences the outcome; lack of blinding is less of a concern. Retention is the outcome and was obtained for all but two trials therefore, attrition and selective outcome reporting bias are probably unimportant. Although the retention trials were fairly well conducted, this could be improved, and they were often poorly reported. This may be because they were designed when loss to follow-up became a problem in a trial, rather than pre-planned prior to the start of the host trial.

Few trials are available for behavioural, case management and methodological strategies (only 1 or 2 each) and this affects the power of the result for these strategies. The use of open trials to increase questionnaire response can only be applied to trials where blinding is not required; based on our result, this strategy would need to be evaluated in different trial contexts if it were to be applied in other areas. All included trials were conducted in higher income countries. Therefore, the effective strategies may not be socially, culturally or economically appropriate to trials conducted in low-resource settings. The diversity of strategies and the low

number of trials meant that we could not examine the impact of, for example, trial setting and disease area as planned. Moreover, most of the evidence relates to increasing questionnaire response rather than participant retention in follow-up. Many trials require participants to return to sites for follow-up and monitoring; however, barriers to follow-up do exist and are trial and participant specific depending on the disease area, treatment and population group. Return for follow-up at sites depends on participant preferences and the demands of the trial.[42] Barriers to follow-up at site could be alleviated by using tailored strategies to encourage participants to return to sites for follow-up and monitoring. Studies that evaluate such strategies are particularly needed.

Edwards' extensive review of methods to increase response to postal and electronic questionnaires found that monetary incentives and recorded delivery of questionnaires improved response.[7] However, unlike our review, they also found that non-monetary incentives, shorter questionnaires, use of handwritten addresses, stamped return envelopes (as opposed to franked return envelopes) and first class outward mailing were effective. We did, however, find that a 'package' including an enhanced letter with several reminders was effective. The trials included in the Edwards' review were embedded in surveys, cohort studies and trials, and there was substantial heterogeneity in the results, which was not a particular problem in this review.[7] Moreover, we included 8 unpublished trials and 18 other trials not included by Edwards.[12]

Nakash *et al*'s[43] small systematic review of ways to increase response to postal questionnaires in healthcare was not exclusive to randomised trials. They found reminder letters, telephone contact and short questionnaires increased response to postal questionnaires. There was no evidence that incentives were effective. A systematic review of methods to increase retention in population-based cohort studies had no meta-analysis, but suggested that incentives were associated with increased retention.[6]

Prior to our review, it was not clear which, if any, of these strategies could be extrapolated to randomised trials. We also identified additional strategies that may improve trial questionnaire response or retention, for example, methodological strategies.

## Implications

Although giving monetary incentives upfront seems effective, offering and giving these after receipt of data could be a cost-effective strategy, because those not returning questionnaires would not receive an incentive. The addition of non-monetary incentives, for example, lapel pins and certificates of appreciation, or offers of these, did not increase response or retention, perhaps because these items are not valued by participants. Offers of monetary incentives were also an effective strategy in the context of an online electronic questionnaire,

thus, it would be beneficial for trialists to know which is more effective: an offer of a monetary incentive or an upfront monetary incentive in a head-to-head trial comparison.

The value of incentives used in the UK evaluations ranged from GBP5 to GBP20 and for US-based studies was US\$2 to US\$10. For offers of entries into prize draws, the values were higher, ranging from GBP25 to GBP250 for the UK prize draws and US\$50 for US-based prize draws. The value of monetary incentive should not be so high as to be perceived as payment or coercion for data but more as an appreciation for efforts made by participants. A cost-effectiveness analysis for additional responses gained after incentive strategies were introduced was reported for only some incentive trials.[18 25 29 30 39] As costs increase the cost-benefit associated with incentive strategies would need to be updated if incentives were to be used to improve retention in future trials.

Priority post, enhanced letters (eg, signed by the principal investigator) and different types of additional reminders are used by trialists in current research practice, but these were not found to be effective. The former may not be considered important and too many reminders, over and above standard procedures, could be counterproductive.

Although appearing very effective, the TDM for postal questionnaires could be labour-intensive to implement, expensive and may no longer be applicable to some participant groups, for example, young people used to other modes of communication, or in trials using email, text or online data collection. Recorded delivery could be useful to ensure trial follow-up supplies to reach their intended destination, but careful planning to avoid inconvenience for the participant might be necessary. Open trials to increase questionnaire response can only be used where blinding is not required. This could be counterproductive, however, as unblinded trials can cause biased outcome assessment or loss to follow-up if a participant or clinician has a treatment preference.

Questionnaire length and relevance may need further evaluation as there is only a suggestion that these are effective in the context of randomised trials. Also, telephone follow-up compared with a monetary incentive sent with a questionnaire needs further evaluation possibly with a cost-benefit analysis as both could be expensive in time and human resources. Evaluations of strategies that encourage participants to return to sites for follow-up visits and monitoring are particularly needed because many trials collect outcome data in this way.

Trialists should consider including well thought out and adequately powered evaluations of strategies to increase retention in randomised trials with clear definitions of retention strategies and retention measures. Trialists could incorporate evaluations of strategies to improve retention at the design stage so that power, sample size and funding are taken into account. Retention trials were often poorly reported and trialists should adhere to the consort

guidelines for trial reporting to facilitate the synthesis of results in future methodology reviews.

There is less research on ways to increase return of participants to trial sites for follow-up and on the effectiveness of strategies to retain trial sites in cluster and individual randomised trials. Research in both areas would be very beneficial to trialists. Application of the results of this review would depend on trial setting, population, disease area, budget allowance and follow-up procedures.

## Conclusions

Trialists should consider using monetary incentives and offers of monetary incentives to increase postal and electronic questionnaire response, depending on trial setting, population, disease area, budget and usual follow-up procedures.

Future evaluations of retention strategies in randomised trials should be carefully planned and adequately powered, and the retention strategies and measures of retention should be clearly defined. More research on ways to increase return of participants to sites for follow-up and on ways to retain sites in cluster and individual randomised trials are also needed.

**Author affiliations**
[1]MRC Clinical Trials Unit at UCL, London, UK
[2]Ethnicity and Health, MRC Social and Public Health Sciences Unit, 4 Lilybank Gardens, Glasgow, UK
[3]PRIMENT Clinical Trials Unit, Research Department of Primary Care and Population Health, UCL Medical School, London, UK

**Acknowledgements** The authors would like to thank the following: all authors of included published trials for providing extra unreported data; principal investigators for data on trials in progress or completed and unpublished (Julia Bailey UCL for data for Bailey 1 and Bailey 2; Graeme MacLennan for data for MacLennan; Stephanie Land data for Land) and the coordinators of the UK Clinical Trials Units who responded to our survey with information about ongoing and/or unpublished completed trials. The authors also thank Cara Booker, SPHRU, for search strategy information; Angela Young, Librarian UCL, for assistance with searching databases and Ian White, MRC Bio Statistics Unit Cambridge, for helpful comments on the analysis of multiarm trials. They also acknowledge Shaun Treweek, Mike Clarke, Andy Oxman, Karen Robinson and Anne Eisinga for comments on the review protocol; and Phil Edwards, Elie Akl, Lisa Maguire, Jean Suvan, Shaun Treweek, Mike Clarke and Karen Robinson for comments on the review.

**Contributors** VCB wrote the protocol for the review with comments from JFT, GR, SS, SM, IN and SH. JFT and VCB designed the searches with comments from SH. VCB conducted the searches, screened all abstracts, and full articles of potentially eligible trials. VCB and GR screened potentially eligible trial articles. SS acted as a third reviewer. Data extraction was conducted by VCB and checked by JT. JT designed the analysis plan with VCB. VCB conducted the analysis with advice on interpretation of results from JFT, SS, IN and GR. VCB wrote the first draft of the review with critical comments from all authors.

**Funding** This project was funded by the Medical Research Council Population Health Sciences Research Network grant number PHSRN 30.

**Competing interests** None.

**Provenance and peer review** Not commissioned; externally peer reviewed.

**Data sharing statement** No additional data are available.

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
