## [Reviewer comments · BMJ Open]

Some articles will have been accepted based in part or entirely on reviews undertaken for other BMJ Group journals. These will be reproduced where possible.

ARTICLE DETAILS

TITLE (PROVISIONAL)	Strategies to improve retention in randomised trials: a Cochrane systematic review and meta-analysis
AUTHORS	Brueton, Valerie; Tierney, Jayne; Stenning, Sally; Meredith, Sarah; Harding, Seeromanie; Nazareth, Irwin; Rait, Greta

VERSION 1 - REVIEW

REVIEWER	McKinstry, Brian University of Edinburgh, centre for population Health Sciences
REVIEW RETURNED	18-Aug-2013

GENERAL COMMENTS	This is a comprehensive review of methods to improve retention within randomised controlled trials. The review reveals that most research done on this area centres on questionnaire return with few studies exploring strategies that encourage participants to return to sites for follow-up visits and monitoring. I found this surprising and it is clear that studies such as these are particularly needed. I think I would have liked a little more discussion around this.
--

REVIEWER	Professor Elaine McColl Director Newcastle CTU 4th Floor William Leech Building The Medical School Newcastle University Framlington Place Newcastle upon Tyne NE2 4HH I am the corresponding author of one of the papers included in this review.
REVIEW RETURNED	02-Sep-2013

GENERAL COMMENTS	This is a useful addition to the literature on retention in clinical trials. The following comments are offered in the spirit of constructive criticism. 1 - the abstract would be improved by elaboration of: (a) the primary outcome - retention of trial participants at what time/follow-up point of the 'host' trial; (b) the distinction between 'adding' and 'offering' an incentive; (c) what is meant by 'case management'. 2 - the search strategy used in this review (at least the version used for MEDLINE) should be included as an appendix to the paper, and the start date for the searches should be stated in the section on 'identification of retention trials'.
---

	3 - greater clarity (page 6) is needed on the time points (baseline or follow-up) in the 'host' trial at which the retention strategies were applied, and why the earliest time point was used in those retention trials which reported retention at multiple time points. In terms of attrition biases and study power, it is surely the number retained to the time point that corresponds to the primary outcome (e.g. 12 month follow-up) in the 'host' trial that is the most important. 4 - please clarify how the eight unpublished trials were identified. 5 - if possible please state the chance of winning the prizes (i.e. the expected value of the incentive) in the studies involving a lottery (page 9). If this information was not available in the reports of those retention trials, say so. 6 - please define 'enhanced letters' (page 9) 7 - please expand a little on what the 'motivational behavioural strategies' involved (page 9) 8 - please elaborate on what is meant by 'case management' (page 9), in particular what 'monitoring' and 'advocacy' meant in this context 9 - it is not entirely clear to me how participants in the retention trials might have been aware of the intervention, but not the evaluation thereof (page 10) - please explain 10 - it is not evident to me why a 'shorter condensed' questionnaire could not be 'clear' but this is the implication of the contrasting of 'long and clear questionnaires' with 'shorter condensed questionnaires' (page 13). Please explain 11 - in the implications section, you could mention that none (?) of the retention trials seemed to have considered cost-effectiveness of the interventions, e.g whether the marginal cost of (say) incentives outweighed the marginal benefit in terms of additional participants retained. The ethics of using incentives could also be discussed more fully. 12 - you could give stronger and more directive recommendations for how future retention trials should be designed, conducted and reported, to improve quality and usefulness. You might also highlight particular strategies that are most worthy of evaluation (which might include strategies not tested in the trials reported here).
--	--

REVIEWER	Huang, Chao Cardiff University
REVIEW RETURNED	22-Oct-2013

GENERAL COMMENTS	In this paper the authors restrict their Cochrane systematic review on the strategies to improve retention rates in random controlled trials. This is different from other existing system reviews, which would give some specific information on retention improvement in RCT. Overall the paper is sound but requires a few modifications.
--

	Here are some points which need to be further addressed.  1. In the section of 'Statistical analysis' (page 7), 0.10 is set as the significant level rather than the conventional 0.05 (or 0.01) significant level. Please address the reason. 2. Additionally, if a certain statistical software was mainly implemented for the meta-analysis in this paper, please provide the details (software name, version, et al.) in the context. 3. In the section of 'Statistical analysis' (page 7), it is mentioned that 'risk ratios were pooled using fixed effect models' if there was no substantial heterogeneity. Have the authors considered the random effect models to tackle the heterogeneity? 4. In the section of 'Results' (page 12, '2. Communication strategies'), the information of subgroup heterogeneity was not provided in strategy 2, which was provided both in '1. Incentive strategies' and '3. New questionnaire strategies'. Please add the relevant details into the context. 5. In the section of 'Results' (page 12, '3. New questionnaire strategies'), the authors mentioned 'Although there is modest heterogeneity between the questionnaire subgroups $p=0.11$ (Figure 3), it did not seem reasonable to pool the results based on such different interventions.' The p value (0.11) suggested that there is no heterogeneity. Please modify the terms used here and provide the reason not pooling the results (since no heterogeneity). 6. In the section of 'Results' (page, 13), quite few trials are available for reviews of strategies 4, 5 and 6 (only one or two each), which would affect the power of the result. Please add some comments on this issue, probably in the section of 'Strengths and weakness'.
--	--

VERSION 1 – AUTHOR RESPONSE

Reviewer: Brian McKinstry
University of Edinburgh, centre for population Health Sciences

This is a comprehensive review of methods to improve retention within randomised controlled trials. The review reveals that most research done on this area centres on questionnaire return with few studies exploring strategies that encourage participants to return to sites for follow-up visits and monitoring. I found this surprising and it is clear that studies such as these are particularly needed . I think I would have liked a little more discussion around this.

Many trials require participants to return to sites for follow-up and monitoring; however barriers to follow-up do exist and are trial and participant specific depending on the disease area, treatment and population group. Return for follow-up at sites depends upon participant preferences and the demands of the trial.⁴⁴ Barriers to follow-up at site could be alleviated by using tailored strategies to encourage participants to return to sites for follow-up and monitoring. Studies that evaluate such strategies are particularly needed.

We have added this to the discussion

Professor Elaine McColl
Director Newcastle CTU
4th Floor William Leech Building
The Medical School
Newcastle University

I am the corresponding author of one of the papers included in this review.

This is a useful addition to the literature on retention in clinical trials. The following comments are

offered in the spirit of constructive criticism.

1 - the abstract would be improved by elaboration of: (a) the primary outcome - retention of trial participants at what time/follow-up point of the 'host' trial; (b) the distinction between 'adding' and 'offering' an incentive; (c) what is meant by 'case management'.

The exact time point in the host trial at which the retention strategies were applied was not specified in the retention trial reports. See response to point three below.

'adding' and 'offering' an incentive and case management have been elaborated upon as far as possible within the confines of the abstract word limit

2 - the search strategy used in this review (at least the version used for MEDLINE) should be included as an appendix to the paper, and the start date for the searches should be stated in the section on 'identification of retention trials'.

The MEDLINE search strategy has been added as an appendix.

The start dates for searches have been added to the section on identification of trials as requested.

3 - greater clarity (page 6) is needed on the time points (baseline or follow-up) in the 'host' trial at which the retention strategies were applied, and why the earliest time point was used in those retention trials which reported retention at multiple time points. In terms of attrition biases and study power, it is surely the number retained to the time point that corresponds to the primary outcome (e.g. 12 month follow-up) in the 'host' trial that is the most important.

The exact time point in the host trial at which the retention strategies were applied was not specified in most retention trial reports. Nevertheless, most retention strategies were applied during follow-up for the host trial when loss to follow-up was anticipated or became a problem. For three host trials the retention strategy was applied in further follow-up of trial participants after completion of the host trial. For four host trials the strategy was applied during a randomised pilot phase of the host trial and for one other host trial the retention strategy was applied before the host trial commenced.

We have added this to the manuscript

The earliest time point was used, if the time of primary outcome was not stated, to see the initial impact on retention or response of introducing the strategy.

4 - please clarify how the eight unpublished trials were identified.

The eight unpublished trials were identified through word of mouth, a survey of UK clinical trials units and through reference lists of the relevant literature.

This has been added to the first paragraph of the results.

5 - if possible please state the chance of winning the prizes (i.e. the expected value of the incentive) in the studies involving a lottery (page 9). If this information was not available in the reports of those retention trials, say so.

No information was available from the included trials on the chance of winning the prize.

This has been added to the incentives strategy section of the results

6 - please define 'enhanced letters' (page 9)

Enhanced letters were those with additional information about trial processes or with an extra feature e.g. signed by a principal investigator.

This has been added to the results

7 - please expand a little on what the 'motivational behavioural strategies' involved (page 9)

A behavioural strategy was defined as giving participants information about goal setting and time management to facilitate successful trial completion.

This has been added to the results section under the heading behavioural strategies.

8 - please elaborate on what is meant by 'case management' (page 9), in particular what 'monitoring' and 'advocacy' meant in this context

This strategy involved trial assistants managing participant follow-up by arranging services to enable participants to keep trial follow-up appointments.

This has been added to the results section under the heading case management strategies.

9 - it is not entirely clear to me how participants in the retention trials might have been aware of the intervention, but not the evaluation thereof (page 10) - please explain

If a strategy was evaluated in a host trial e.g. gifts or monetary incentives, the participants may not know that an evaluation of the effectiveness of the strategy was being undertaken unless they were specifically informed that this was the case.

10 - it is not evident to me why a 'shorter condensed' questionnaire could not be 'clear' but this is the implication of the contrasting of 'long and clear questionnaires' with 'shorter condensed questionnaires' (page 13). Please explain

The trial by Subar examined "length and clarity" of questionnaires on response rates. The Dillman approach was used to design a longer and clearer questionnaire that was thought to be cognitively easier for participants to complete, the implication being that short and condensed questionnaires are perhaps more difficult to complete for some participants.

11 - in the implications section, you could mention that none (?) of the retention trials seemed to have considered cost-effectiveness of the interventions, e.g whether the marginal cost of (say) incentives outweighed the marginal benefit in terms of additional participants retained. The ethics of using incentives could also be discussed more fully.

The value of incentives used in UK evaluations ranged from GBP5 to GBP20 and for US-based studies USD2 to USD10. For offers of entries into prize draws, the values were higher, ranging from GBP25 to GBP250 for UK prize draws and USD50 for US-based prize draws. The value of monetary incentive should not be so high as to be perceived as payment or coercion for data but more as an appreciation for efforts made by participants. A cost effectiveness analysis for additional responses gained after incentive strategies were introduced was reported for only some incentive trials. As costs increase the cost benefit associated with incentive strategies would need to be updated if incentives were to be used to improve retention in future trials 25;29;39 18;30.

This has been added to the implications

12 - you could give stronger and more directive recommendations for how future retention trials should be designed, conducted and reported, to improve quality and usefulness. You might also highlight particular strategies that are most worthy of evaluation (which might include strategies not tested in the trials reported here).

Trialists should consider including well thought out and adequately powered evaluations of strategies to increase retention in randomised trials with a clear definition of retention strategies and retention measures. Trialists could incorporate evaluations of strategies to improve retention at the design stage so that power, sample size and funding are taken into account. Retention trials were often poorly reported and trialists should adhere to the consort guidelines for trial reporting to facilitate the synthesis of results in future methodology reviews.

There is less research on ways to increase return of participants to trial sites for follow-up and on the effectiveness of strategies to retain trial sites in cluster and individual randomised trials. Research in both areas would be very beneficial to trialists. Application of the results of this review would depend on trial setting, population, disease area, budget allowance and follow-up procedures.

This has been added to the implications

Reviewer: Chao huang
Cardiff University

If you have any further comments for the authors please enter them below.

In this paper the authors restrict their Cochrane systematic review on the strategies to improve

retention rates in random controlled trials. This is different from other existing system reviews, which would give some specific information on retention improvement in RCT. Overall the paper is sound but requires a few modifications.

Here are some points which need to be further addressed.

1. In the section of 'Statistical analysis' (page 7), 0.10 is set as the significant level rather than the conventional 0.05 (or 0.01) significant level. Please address the reason.

0.10 is set as the significance level rather than the conventional 0.05 because the Chi2 test for heterogeneity has low power and this is recommended by the Cochrane handbook.

2. Additionally, if a certain statistical software was mainly implemented for the meta-analysis in this paper, please provide the details (software name, version, et al.) in the context.

RevMan5 was used for all of our statistical analyses.

This has been added to the statistical analysis section of the paper.

3. In the section of 'Statistical analysis' (page 7), it is mentioned that 'risk ratios were pooled using fixed effect models' if there was no substantial heterogeneity. Have the authors considered the random effect models to tackle the heterogeneity?

We tried wherever possible to explore heterogeneity using subgroup analysis instead but where we have not been able to explain heterogeneity, we have added the results of the subgroup analyses to assess the robustness of the results to the choice of model.

In the non-monetary incentive results we have include the random effects results alongside the fixed effects and added the following "unless otherwise stated results from the random effects model were similar".

We have also added the following to the statistical analysis section. If heterogeneity could not be explained we used the random effects model to assess the robustness of the results to the choice of model.

4. In the section of 'Results' (page 12, '2. Communication strategies'), the information of subgroup heterogeneity was not provided in strategy 2, which was provided both in '1. Incentive strategies' and '3. New questionnaire strategies'. Please add the relevant details into the context.

Communication strategies were so diverse that these were analysed separately e.g. enhanced letters vs standard letters, total design method versus customary method for follow-up, priority versus regular post. This has been added to the communication section of the results.

5. In the section of 'Results' (page 12, '3. New questionnaire strategies'), the authors mentioned 'Although there is modest heterogeneity between the questionnaire subgroups $p=0.11$ (Figure 3), it did not seem reasonable to pool the results based on such different interventions.' The p value (0.11) suggested that there is no heterogeneity. Please modify the terms used here and provide the reason not pooling the results (since no heterogeneity).

Although there was only some heterogeneity between the questionnaire subgroups (P value = 0.11) , it did not seem reasonable to pool the results based on such different interventions. We have modified the terms used in the results section under New questionnaire strategies.

6. In the section of 'Results' (page, 13), quite few trials are available for reviews of strategies 4, 5 and 6 (only one or two each), which would affect the power of the result. Please add some comments on this issue, probably in the section of 'Strengths and weakness'.

We agree that as few trials were available for the behavioural, case management and methodological strategies (only one or two each),this affects the power of the result for these strategies. The use of open trials to increase questionnaire response can only be applied to trials where blinding is not required however, based on our result this strategy would need to be evaluated in different trial context if it were to be applied in other areas.

We have added a sentence about this in the strengths and weakness section of the discussion